# A proteomic-informed view of the changes induced by loss of cellular adherence: The example of mouse macrophages

**Sacnite Ramirez Rios**[1ʘ], **Anaelle Torres**[2ʘ], **Hélène Diemer**[3,4], **Véronique Collin-Faure**[2], **Sarah Cianférani**[3,4], **Laurence Lafanechère**[1], **Thierry Rabilloud**[2]*

**1** Institute for Advanced Biosciences, Univ. Grenoble Alpes, CNRS UMR 5309, INSERM U1209, Grenoble, France, **2** Chemistry and Biology of Metals, Univ. Grenoble Alpes, CNRS UMR5249, CEA, IRIG-DIESE-CBM-ProMD, Grenoble, France, **3** Laboratoire de Spectrométrie de Masse BioOrganique (LSMBO), Université de Strasbourg, CNRS, IPHC UMR 7178, Strasbourg, France, **4** Infrastructure Nationale de Protéomique, FR2048 ProFI, Strasbourg, France

ʘ These authors contributed equally to this work.
* thierry.rabilloud@cnrs.fr

**Data Availability Statement:** All relevant data are within the manuscript and its supporting information.

## Abstract

Except cells circulating in the bloodstream, most cells in vertebrates are adherent. Studying the repercussions of adherence *per se* in cell physiology is thus very difficult to carry out, although it plays an important role in cancer biology, e.g. in the metastasis process. In order to study how adherence impacts major cell functions, we used a murine macrophage cell line. Opposite to the monocyte/macrophage system, where adherence is associated with the acquisition of differentiated functions, these cells can be grown in both adherent or suspension conditions without altering their differentiated functions (phagocytosis and inflammation signaling). We used a proteomic approach to cover a large panel of proteins potentially modified by the adherence status. Targeted experiments were carried out to validate the proteomic results, e.g. on metabolic enzymes, mitochondrial and cytoskeletal proteins. The mitochondrial activity was increased in non-adherent cells compared with adherent cells, without differences in glucose consumption. Concerning the cytoskeleton, a rearrangement of the actin organization (filopodia *vs* sub-cortical network) and of the microtubule network were observed between adherent and non-adherent cells. Taken together, these data show the mechanisms at play for the modification of the cytoskeleton and also modifications of the metabolic activity between adherent and non-adherent cells.

## Introduction

In vertebrates, all cells except circulating blood cells must adhere to support their normal growth and functions. The adherence to extracellular matrix and/or other cells is critical and adherent cells placed in non-adherent conditions either die or form multicellular spheroids. Placing cells in non-adherent conditions has been used to induce differentiation in teratocarcinoma cells [1–3] and more recently to form organoids (e.g. in [4, 5]). Because of such

**Funding:** This work was supported by Agence Nationale pour la Recherche, France in the form of a grant awarded to SC (ANR-10-INBS-08-03), as well as University Grenoble Alpes and French National Centre for Scientific Research (CNRS) in the form of recurring basic funding for the labs and research teams. The funders had no role in study design, data collection and analysis, decision to publish, or preparation of the manuscript.

**Competing interests:** The authors have decalred no competing interests.

important consequences induced by cell adhesion on cell growth and function, the transition between adherent and non-adherent states is rather rare. There are however physiological situations, such as blood cells diapedesis, during which cells that circulate into the blood stream must adhere to the endothelial cells and cross the endothelial barrier to reach target tissues.

Another example of transition, from an adherent to a non-adherent state, is observed in the metastatic process, where cells detach from the tumor mass and circulate in the blood and lymphatic vasculature prior to reattaching and extravasating to colonize distant organs [6].

The comparative analysis of the only effects of adherence on cellular functions is complicated by the fact that in many studies models the acquisition or loss of adherence induces major alterations in cell physiology that would obscure the effects of the adherence itself. For example, P19 teratocarcinoma cells differentiate in suspension spheroids while they do not in adhering conditions [7]. In this context, the comparison between spheroids and adherent cells would not be a comparison between adherent and non-adherent cells, but between differentiated cells adhering between them and undifferentiated cells adhering on plastic.

Mouse macrophage cell lines represent one of the rare experimental models that may be suitable to compare the adherent and non-adherent states. Indeed, they grow equally well under adherent and non-adherent conditions and keep their differentiated functions under both conditions. We therefore decided to use this model to analyze the changes between the adherent and the non-adherent state using a broad approach, based on proteomics.

## Materials and methods

Unless specified otherwise, the chemicals used in this work were purchased from Sigma-Millipore and were at least 99% pure.

### Cell culture

The mouse monocyte/macrophage cell line RAW264.7 was purchased from the European Cell Culture Collection (Salisbury, UK). The cells were cultured in RPMI 1640 medium supplemented with 10% fetal bovine serum. Because the cells are strongly adherent to classical culture plastics, they are easily damaged when passaged, either with trypsin digestion, chemical detachment with EDTA or with scraping. They were thus routinely cultured on non-treated plastics for non-adherent cells, from which they were easily removed for passaging. Cells were seeded every two days at 200,000 cells/mL and harvested at 1,000,000 cells per ml. For the adherent vs. suspension cultures, cells were seeded at 500,000 cells/ml on either adherent T75 flasks or 6-well plates (Corning) or on non-adherent T75 flasks or 6-well plates (suspension culture flasks from Greiner) and let to grow for 24 hours. For harvesting the adherent cells in the adherent flasks, the culture medium (containing a few non-adherent cells) was removed, and the adherent cell layer was rinsed 3 times with serum-free RPMI 1640 medium. The cells were then scraped in Hepes buffered saline (Hepes-NaOH pH 7.5 10mM, NaCl 150mM, $MgCl_2$ 2mM) and collected by centrifugation (400g, 5 minutes). For harvesting the non-adherent cells in the non-adherent flasks, the flasks were shaken and the cell medium containing the non-adherent cells was collected, allowing the few adherent cells to stay within the culture flask. The suspension was centrifuged (400g, 5 minutes) and the cell pellet was rinsed 3 times with serum-free RPMI 1640 medium and once with Hepes buffered saline. The cell pellets were then processed for further use.

For harvesting cells from adherent plates, the culture medium was removed, and the adherent cell layer was rinsed once with serum-free RPMI 1640 medium. The cells were then detached by incubation in PBS containing 1mM EDTA for 5 minutes, following by flushing repeatedly the cell layer. The cell suspension was then diluted with an equal volume of serum-

free RPMI 1640 medium, and the cells were collected by centrifugation. For harvesting the non-adherent cells in the non-adherent plates, repeated flushing was used. The suspension was centrifuged (400g, 5 minutes) and the cell pellet was rinsed 3 times with serum-free RPMI 1640 medium.

## Adherence tests and cell cycle analysis

For the adherence test, cells were seeded at 500,000 cells/ml on either adherent T25 flasks (Corning) or on non-adherent T25 flasks (suspension culture flasks from Greiner) and let to grow for 24 hours. The culture medium was then recovered and the number of non-adherent cells was determined by cell counting.

For the cell cycle analysis, the cells were cultured on adherent or non-adherent plates, as described above. To avoid any bias that may be due to selective damage of adherent cells in any phase of the cell cycle, the analysis was carried out on isolated nuclei rather than on detached cells. To this purpose, a protocol based on the citric acid method described by Miller [8] was used. Cells were seeded at 500,000 cells/ml on 6-well plates (Corning) or on non-adherent 6-well plates (suspension culture flasks from Greiner) and let to grow for 24 hours. For adherent cells, the culture medium was removed, and the cell layer rinsed twice with PBS. The cell layer was the lysed by addition of 0.5 ml of ice-cold fractionation buffer (25 mM Hepes pH 7.5, 25 mM NaCl, 10 mM MgCl2, 0.5% Triton X100). After 1 minute of lysis, 25 µl of 20% (w/v) citric acid were added, and the plate was swirled for another 2 minutes. The suspension was then recovered and centrifuged (800g, 2 minutes, 4˚C). the nuclei pellet was then resuspended in 500µl PBS containing 5 µl RNAse A (1mg/ml) and 5µl propidium iodide (1mg/ml).

For non-adherent cells, the cells were first collected by centrifugation (400g, 5 minutes), then washed twice with PBS. The cell pellet was then resuspended in 0.5 ml of ice-cold fractionation buffer and lysed for 1 minute. After that time, 25 µl of 20% (w/v) citric acid were added and mixed by pipetting, and the lysis was allowed to proceed for another 2 minutes. The nuclei were recovered and treated as described above for adherent cells.

Cell cycle analysis was carried out on the labelled nuclei by flow cytometry. A FacsCalibur cytometer (Beckton Dickinson) was used with the CellQuest Pro software. The data were obtained using the following parameters: linear acquisition of the events, doublets eliminated by gating on singlet events and the mean fluorescent intensity of the propidium iodide was represented on a linear mode to identify the different cell cycle stages.

## Proteomics

The 2D gel based proteomic experiments were essentially carried out as previously described [9], on independent biological quadruplicates.

Briefly, the spots selected for identification were excised from silver-stained gels and destained with ferricyanide/thiosulfate on the same day as silver staining in order to improve the efficiency of the identification process [10, 11]. In gel digestion was performed with an automated protein digestion system, MassPrep Station (Waters, Milford, USA). The gel plugs were washed twice with 50 µL of 25 mM ammonium hydrogen carbonate ($NH_4HCO_3$) and 50 µL of acetonitrile. The cysteine residues were reduced by 50 µL of 10 mM dithiothreitol at 57˚C and alkylated by 50 µL of 55 mM iodoacetamide. After dehydration with acetonitrile, the proteins were cleaved in gel with 10 µL of 12.5 ng/µL of modified porcine trypsin (Promega, Madison, WI, USA) in 25 mM $NH_4HCO_3$. The digestion was performed overnight at room temperature. The generated peptides were extracted with 30 µL of 60% acetonitrile in 0.1% formic acid. Acetonitrile was evaporated under vacuum before nanoLC-MS/MS analysis.

NanoLC-MS/MS analysis was performed using a nanoACQUITY Ultra-Performance-LC (Waters Corporation, Milford, USA) coupled to a TripleTOF 5600 (Sciex, Ontario, Canada) mass spectrometer. Mass calibration of the analyser was achieved using peptides from digested BSA. The complete system was fully controlled by AnalystTF 1.7 (Sciex). Raw data collected were processed and converted with MSDataConverter in.mgf peak list format.

For protein identification, the MS/MS data were interpreted using a local Mascot server with MASCOT 2.6.2. algorithm (Matrix Science, London, UK) against UniProtKB/SwissProt (version 2020–04, 563,082 sequences). The protein identification search was carried out in all species. Spectra were searched with a mass tolerance of 15 ppm for MS and 0.05 Da for MS/MS data, allowing a maximum of one trypsin missed cleavage. Carbamido-methylation of cysteine residues, oxidation of methionine residues, acetylation of protein N-terminus, phosphorylation of serine, threonine and tyrosine residues were specified as variable modifications. Protein identifications were validated with at least two peptides with Mascot ion score above 30.

Mass spectrometry data are available via ProteomeXchange with identifier PXD021593.

## Enzyme assays

The enzymes activities were assayed according to published procedures (see below).

The cell extracts for enzyme assays were prepared by lysing the cells for 20 minutes at 0°C in 20 mM Hepes pH 7.5, 2 mM $MgCl_2$, 50 mM KCl, 1 mM EGTA, 0.15% (w/v) tetradecyldi-methylammonio propane sulfonate (SB 3–14), followed by centrifugation at 15,000 g for 15 minutes to clear the extract. The protein concentration was determined by a dye-binding assay [12]. The dehydrogenases or dehydrogenases-coupled activities were assayed at 500nm using the phenazime methosulfate/iodonitrotetrazolium coupled assay [13, 14]. The enzyme assay buffer contained 25mM Hepes NaOH pH 7.5, 5mM magnesium acetate, 100mM potassium chloride and 1%(w/v) Triton X-100. It also contained 30μM phenazine methosulfate, 200μM iodonitrotetrazolium chloride, 250μM of the adequate cofactor (NAD or NADP) and 1-5mM of the organic substrate, which was used to start the reaction. Triose phosphate isomerase was assayed with dihydroxyacetone phosphate and a glyceraldehyde dehydrogenase-coupled assay [15]. Hexokinase was assayed by a glucose 6-phosphate dehydrogenase (G6PDH)-coupled assay [16]. Enolase was assayed at 340nm by a pyruvate kinase-lactate dehydrogenase-coupled assay [17]. Phosphoglycerate mutase was assayed by measuring the decrease of NADH at 340 nm in a pyruvate kinase-lactate dehydrogenase-coupled assay [18]. Pyruvate kinase was assayed by measuring the decrease of NADH at 340 nm in a lactate dehydrogenase-coupled assay [19].

## Mitochondrial transmembrane potential measurement

The mitochondrial transmembrane potential was assessed by Rhodamine 123 uptake. Cells were incubated with Rhodamine 123 (80 nM) for 30 minutes at 37°C [20], 5% CO2 then rinsed twice in cold Glucose (1 mg/mL)—PBS (PBSG) and harvested in cold PBSG supplemented with Propidium Iodide (1 μg/mL). The mitochondrial potential of cells was analyzed by flow cytometry on a FacsCalibur instrument (Beckton Dickinson). The dead cells (propidium positive) were excluded of the analysis. The low rhodamine concentration was used to avoid intra-mitochondrial fluorescence quenching that would result in a poor estimation of the mitochondrial potential [21].

The glucose concentration in conditioned media was determined using a clinical glucometer and measuring the residual glucose concentration at the end of the culture period and in the starting culture medium.

## Phagocytosis and nitric oxide production assay

The phagocytic capacity of the cells was measured using fluorescent latex beads and flow cytometry [22, 23]. Both the proportion of phagocytic cells and the mean fluorescence, giving an index of the number of phagocytosed beads, were determined. Nitric oxide production induced by lipopolysaccharide (LPS) stimulation was determined using the Griess reagent, as previously described [23], and corrected by the number of cells in the wells. For cytokine production, a commercial kit (BD Cytometric Bead Array, catalog number 552364 from BD Biosciences) was used.

## Microscopic analysis of the cytoskeleton

**Antibodies.** For microtubule cytoskeleton analysis the primary antibody used was the anti α-tubulin (clone α3A1) produced by L. Lafanechère [24]. Actin microfilaments were visualized using phalloidin-rhodamine (Sigma, P1951). DNA was stained with 20 μM Hoechst 33342 (Sigma, #23491-52-3).

**Immunofluorescence.** For immunofluorescence analysis cells were seeded at 300 000 cells/ml on 12-well plates in adherent or suspension conditions (see above). After 24 hours cells were fixed with 3.7% PFA (Sigma) for 20 min, permeabilized with 0.1% Triton in PBS (Sigma), then washed with PBS, and blocked with a blocking solution (3% BSA/10% goat serum/PBS) for 1 hr. Samples were incubated for 2hours at room temperature with the primary antibody and phalloidin diluted in blocking solution, followed by three washes with PBS containing 0.2% Tween20. The cells were then incubated with secondary antibody at room temperature for 1 hr followed by three washes with PBS containing 0.2% Tween20. In the case of non-adherent/suspension cells, samples were centrifuged at 1200 rpm for 4 minutes between each step. Samples were finally mounted using mounting Mowiol medium. Images were captured with a Zeiss AxioimagerM2 microscope equipped with the acquisition software AxioVision. Contrast of the colored images was adjusted using the ImageJ software.

## Results

### Adherence and cell cycle

As cells were routinely cultured on non-adherent flasks, we first determined which percentage of cells were adherent or non-adherent when transferred to the corresponding culture substrates. When cells were seeded at 500,000 cells/ml and harvested 24 hours later, 855,555 ±156,741 cells/ml (range 688,888–1,000,000) were recovered as non-adherent from non-adherent flasks (indicating cell growth) whereas 29,629±27962 cells/ml (range 0–55,555) were recovered as non-adherent from adherent flasks. Even if assuming no cell growth, this means that more that 94% of the cells adhered on classical culture plastics.

In order to investigate if cell proliferation could be different under both conditions, the cell cycle was analyzed and the results are presented in Table 1.

These data showed that the proportion of cells engaged in cycling (S and G2/M) was not significantly different between adherent and non-adherent cells.

**Table 1. Cell cycle of non-adherent and adherent cells.**

|  | Cell cycle stages | | | |
|---|---|---|---|---|
|  | **% Sub G1** | **% G1** | **% S** | **% G2/M** |
| Non adherent | 1.4 ± 0.1 | 59.9 ± 0.5 | 15.8 ± 0.1 | 20.6 ± 0.8 |
| Adherent | 3.2 ± 0.2 | 55.3 ± 0.5 | 16.8 ± 0.6 | 21.7 ± 0.8 |

## Proteomics

Adherent and non-adherent cells extracts were prepared and their protein content was separated using two-dimensional gel electrophoresis. The raw images used for the analysis are shown in S1 and S2 Figs. Proteins were then identified using quantitative proteomics.

Spots of interest were selected on the two-dimensional gel images on the basis of statistical tests, using the numerical data provided by the 2D gel analysis software (S1 Table). Selected spots showed either a U value of 0 in the Mann-Whitney test (corresponding to a p value ≤ 0.028) or a U value of 1 (corresponding to a p value ≤0.058 and a p value 0.05 in the Welch test). Of the 198 spots that met that criteria, 111 were identified by mass spectrometry, and the spot statistical and identification parameters are shown in S2 Table. Annotated gel images showing these proteins are shown in Fig 1 and S3 Fig. The spot list was then processed using the David software tool [25, 26] to highlight modulated pathways, and the results are displayed in S3 Table.

As expected, "cell adhesion" appeared in the top clusters, but other pathways were also highlighted, which prompted us to validate them. A selection of such proteins is presented in Table 2.

Furthermore, 2D gel-based proteomics allows the visualization of unmodified- and post-translationally modified protein forms. Our data indeed often showed proteins appearing as

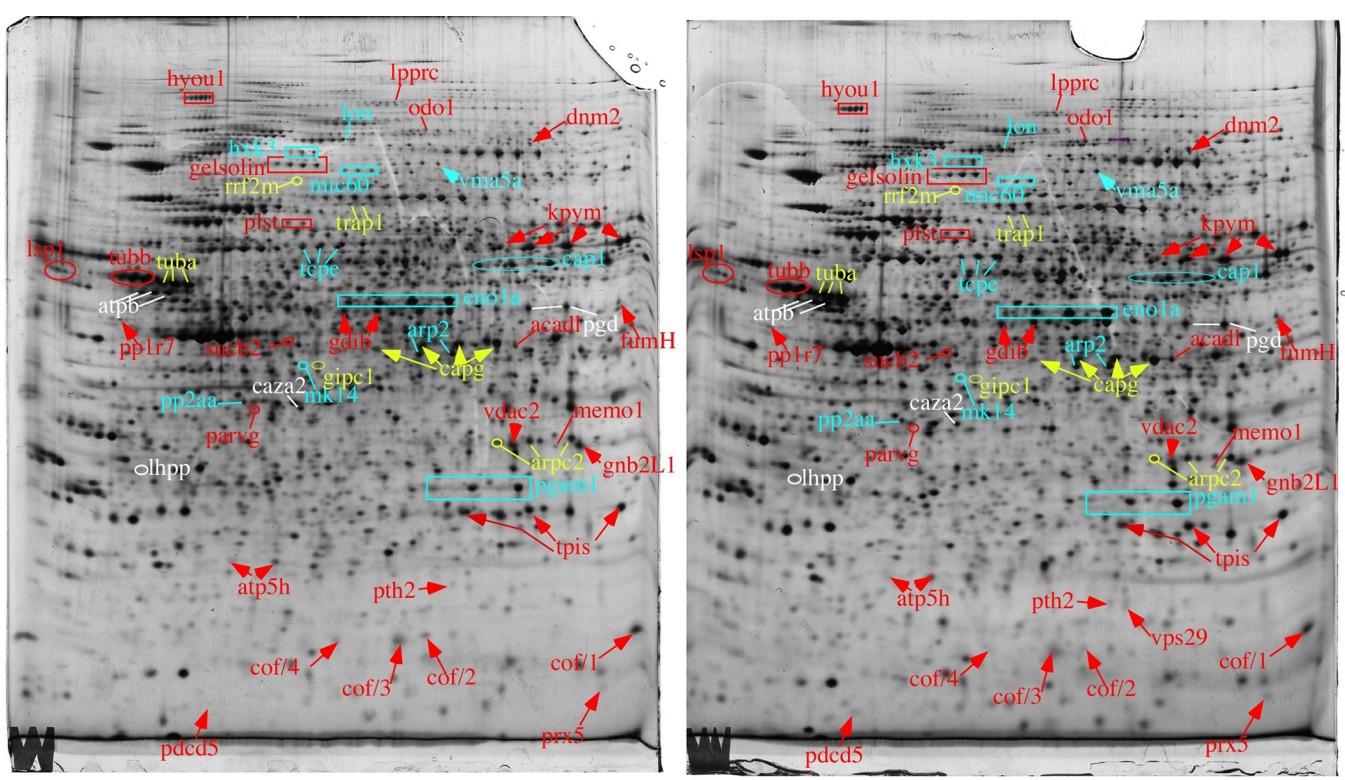

Adherent                    Suspension

**Fig 1. Proteomic analysis of total cell extracts by 2D electrophoresis.** Total cell extracts of RAW264.7 cells were separated by two-dimensional gel electrophoresis. The first dimension covered a 4–8 pH range and the second dimension a 15–200 kDa range. Total cellular proteins (150 μg) were loaded on the first-dimension gel. The proteins highlighted on the figure show changes between the adherent and suspension conditions, for at least one spot in case of proteins showing multiple spots. Only the proteins belonging to the categories: "associated with cytoskeleton", "associated with signal transduction", "associated with tumors", "central metabolism" and "mitochondrial" are shown on this figure. The other proteins are shown in S3 Fig.

**Table 2. Selection of proteins changing in abundance between adherent and non-adherent cells.**

| Short name (spot code) | Uniprot accession number | Protein name | Condition of highest abundance |
|---|---|---|---|
| **Cytoskeleton-associated proteins** | | | |
| arp2/2 | P61161 | Actin-related protein 2 | A |
| arpc2/3 | Q9CVB6 | Actin-related protein 2/3 complex subunit 2 | A |
| capg/2 | P24452 | Macrophage-capping protein | A |
| capg/3 | P24452 | Macrophage-capping protein | A |
| caza2 | P47754 | F-actin-capping protein subunit alpha-2 | A |
| cof1/2 | P18760 | Cofilin-1 | A |
| cof1/4 | P18760 | Cofilin-1 | A |
| dnm2 | P39054 | Dynamin-2 | NA |
| gdib/1 | Q61598 | Rab GDP dissociation inhibitor beta | A |
| gelsolin/1 | P13020 | Gelsolin | NA |
| gipc1 | Q9Z0G0 | PDZ domain-containing protein GIPC1 | NA |
| lsp1/1 | P19973 | Lymphocyte-specific protein 1 | A |
| lsp1/2 | P19973 | Lymphocyte-specific protein 1 | A |
| memo1 | Q91VH6 | Protein MEMO1 | A |
| parvg | Q9ERD8 | Parvin gamma | A |
| plst/1 | Q61233 | Plastin-2 | NA |
| plst/2 | Q61233 | Plastin-2 | NA |
| tcpe/2 | P80316 | T-complex protein 1 subunit epsilon | A |
| tubb/2 | P99024 | Tubulin beta | A |
| tubb/3 | P99024 | Tubulin beta | A |
| **Central metabolism enzymes** | | | |
| eno1/3 | P17182 | Alpha-enolase | A |
| kpym/2 | P52480 | pyruvate kinase | A |
| hxk3/2 | Q3TRM8 | Hexokinase-3 | NA |
| pgam/2 | Q9DBJ1 | Phosphoglycerate mutase 1 | A |
| pgam/3 | Q9DBJ1 | Phosphoglycerate mutase 1 | A |
| pgd/2 | Q9DCD0 | 6-phosphogluconate dehydrogenase, decarboxylating | A |
| tpis/3 | P17751 | Triosephosphate isomerase | A |
| **Detoxification proteins** | | | |
| aldr/2 | P45376 | Aldose reductase | A |
| prx1 ox | P3570 | Peroxiredoxin-1 | A |
| prx3* | P20108 | Peroxiredoxin-3 | NA |
| prx5* | P99029 | Peroxiredoxin-5 | NA |
| **Annexins** | | | |
| anxa1/3 | P10107 | Annexin A1 | A |
| anxa3/3 | O35639 | Annexin A3 | A |
| anxa4/2 | P97429 | Annexin A4 | A |
| **Mitochondrial proteins** | | | |
| atp5h/2 | Q9DCX2 | ATP synthase subunit d, mitochondrial | A |
| atpb/2 | P56480 | ATP synthase subunit beta, mitochondrial | A |
| atpb/3 | P56480 | ATP synthase subunit beta, mitochondrial | A |
| acadl | P51174 | Long-chain specific acyl-CoA dehydrogenase | NA |
| fumh | P97807 | Fumarate hydratase, mitochondrial | A |
| lon | Q8CGK3 | Lon protease homolog, mitochondrial | NA |

(*Continued*)

**Table 2.** (Continued)

| Short name (spot code) | Uniprot accession number | Protein name | Condition of highest abundance |
|---|---|---|---|
| lpprc | Q6PB66 | Leucine-rich PPR motif-containing protein, mitochondrial | NA |
| mic60/3 | Q8CAQ8 | MICOS complex subunit Mic60 | NA |
| odo1 | Q60597 | 2-oxoglutarate dehydrogenase, mitochondrial | NA |
| pth2 | Q8R2Y8 | Peptidyl-tRNA hydrolase 2, mitochondrial | NA |
| rrf2m | Q8R2Q4 | Ribosome-releasing factor 2, mitochondrial | NA |
| sucb2 | Q9Z2I8 | Succinyl-CoA ligase [GDP-forming] subunit beta, mitochondrial | NA |
| trap1/1 | Q9CQN1 | Heat shock protein 75 kDa, mitochondrial | |
| vdac2 | Q60930 | Voltage-dependent anion-selective channel protein 2 | NA |
| **Associated with tumors** | | | |
| gipc1 | Q9Z0G0 | PDZ domain-containing protein GIPC1 | NA |
| hyou1/1 | Q9JKR6 | Hypoxia up-regulated protein 1 | NA |
| pdcd5 | P56812 | Programmed cell death protein 5 | NA |
| vma5a | Q99KC8 | von Willebrand factor A domain-containing protein 5A | NA |
| **Proteasome/ubiquitin** | | | |
| atg3 | Q9CPX6 | Ubiquitin-like-conjugating enzyme ATG3 | NA |
| brcc3 | P46737 | Lys-63-specific deubiquitinase BRCC36 | NA |
| mindy3 | Q9CV28 | Protein FAM188A | NA |
| psb4/2 | P99026 | Proteasome subunit beta type-4 | A |
| psd13 | Q9WVJ2 | 26S proteasome non-ATPase regulatory subunit 13 | NA |
| **Associated with signal transduction** | | | |
| gnb2L1 | P68040 | Guanine nucleotide-binding protein subunit beta-2-like 1 | A |
| lhpp | Q9D7I5 | Phospholysine phosphohistidine inorganic pyrophosphate phosphatase | NA |
| mk14 | P47811 | Mitogen-activated protein kinase 14 | NA |
| pp1R7 | Q3UM45 | Protein phosphatase 1 regulatory subunit 7 | A |
| pp2aa | P63330 | Serine/threonine-protein phosphatase 2A catalytic subunit alpha isoform | A |

A: adherent cells; NA: non-adherent cells

*: prx 3 and prx5 are also mitochondrial proteins.

spots. We noticed differences between the changes of abundance of some spots in such series, when comparing between the two conditions tested, as exemplified in Fig 1. We therefore decided to reanalyze the MS/MS spectra to detect modified peptides, and especially phosphorylations and acetylations. Phosphorylations were detected on modified forms of capg (T3), annexin 1 (T101; T114; T169; S244) and annexin 3 (T106/108; S139). To our knowledge, none of these phosphorylations has been described yet. Regarding cofilin, for which the phosphorylation landscape is easier to investigate [27], we could compare by spectral counts the degree of phosphorylation on different sites. The results, displayed in S4 Table, showed a higher phosphorylation level for adherent cells, not only on the well-known S3 site [28], but also on less well known sites such as S23/24 [29], T 63 and Y82 [27].

## Enzyme activities

The comparison of adherent cells with non-adherent cells using this proteomic approach highlighted abundance changes in protein forms for several metabolic enzymes, such as acyl-CoA dehydrogenase (acadl P51174), hexokinase (hxk3 Q3TRM8), phosphoglycerate mutase

**Table 3. Enzyme activities.**

|  | adherent | non adherent | ratio | T-test | U-test |
|---|---|---|---|---|---|
| long chain acylCoA DH | 5.78±1.54 | 9.28±0.96 | 1.61 | 0.011 | 0 |
| enolase | 218.2±16.6 | 201.2±17.11 | 0.92 | 0.2 | 3 |
| hexokinase | 8.11±0.57 | 10.11±0.79 | 1.25 | 0.0078 | 0 |
| phosphoglycerate mutase | 1004±117 | 808±60 | 0.80 | 0.036 | 0 |
| pyruvate kinase | 1737±43 | 1635±157 | 0.94 | 0.29 | 4 |
| triose phosphate isomerase | 91.6±7.9 | 112±6.2 | 1.22 | 0.0074 | 0 |

All the activities are expressed in nmole substrate converted/min/mg total protein.

(PGAM Q9DBJ1), pyruvate kinase (KPYM P52480) or triose phosphate isomerase (TPIS P17751). Indeed, the pathway "carbon metabolism" appeared in the pathway analyses. We thus tested the activity of some of these enzymes to validate the proteomic findings. The results, displayed on Table 3, showed a good agreement between the proteomic data and the activity data, with a concomitant increase in hexokinase and acyl-CoA dehydrogenase when comparing the adherent state to the non-adherent one, and a concomitant decrease for phosphoglycerate mutase. Regarding the enolase and pyruvate kinase activities, the general trends were similar between the proteomic and activity data, although the changes in activity were not statistically significant. For triose phosphate isomerase, the only spot changing in abundance was decreased while the activity increased. These discrepancies indicate that the relationship between activity and modification profiles can be complex. For example, phosphorylation can increase activity (e.g. biliverdin reductase [30]) but also decrease it (e.g. pyruvate dehydrogenase [31]). The same unpredictable trend also holds true for acetylation [32].

## Mitochondrial transmembrane potential

Several mitochondrial proteins were highlighted in the proteomic screen, such as trap1 (Q9CQN1), odo1 (Q60597), mic60 (Q8CAQ8), lon (Q8CGK3), lpprc (Q6PB66), peroxiredoxins 3 (P20108) and 5 (P99029), succinylCoA ligase (sucb2 Q9Z2I8), and the "mitochondrion" cluster appeared in the pathway analysis. Furthermore, most of the mitochondrial proteins detected as changing showed an increase in their relative abundance in the non-adherent state compared to the adherent one, leading to the hypothesis that the mitochondrial function may be increased in non-adherent cells. To test this hypothesis, we measured the mitochondrial transmembrane potential. The results showed that the proportion of rhodamine-positive cells (i.e. viable, metabolically active cells) was high and similar under both conditions (99.8±0.06% for adherent cells vs. 99.7±0.35 for non-adherent cells). However, the rhodamine fluorescence intensity, which is a surrogate marker of the mitochondrial transmembrane potential [21], was significantly increased in the non-adherent condition (3844±335 fluorescence units) compared to the adherent one (2571±103 fluorescence units) suggesting an increased mitochondrial activity in the non-adherent cells. In addition, GIPC1, which is one the proteins highlighted in the proteomic screen, interacts with the glucose transporter GLUT1 [33]. Consequently, there could be a different glucose usage between adherent and non-adherent cells.

We thus measured the glucose consumption of the cells after 24 hours in culture. At an equal seeding density of 500,000 cells/ml, adherent cells consumed 0.29±0.017 g/l glucose while non-adherent cells consumed 0.293±0.015 g/l glucose, i.e. equivalent amounts. Thus, the increased mitochondrial activity was not just due to an overall greater energy consumption, but represented a modification induced by adherence loss.

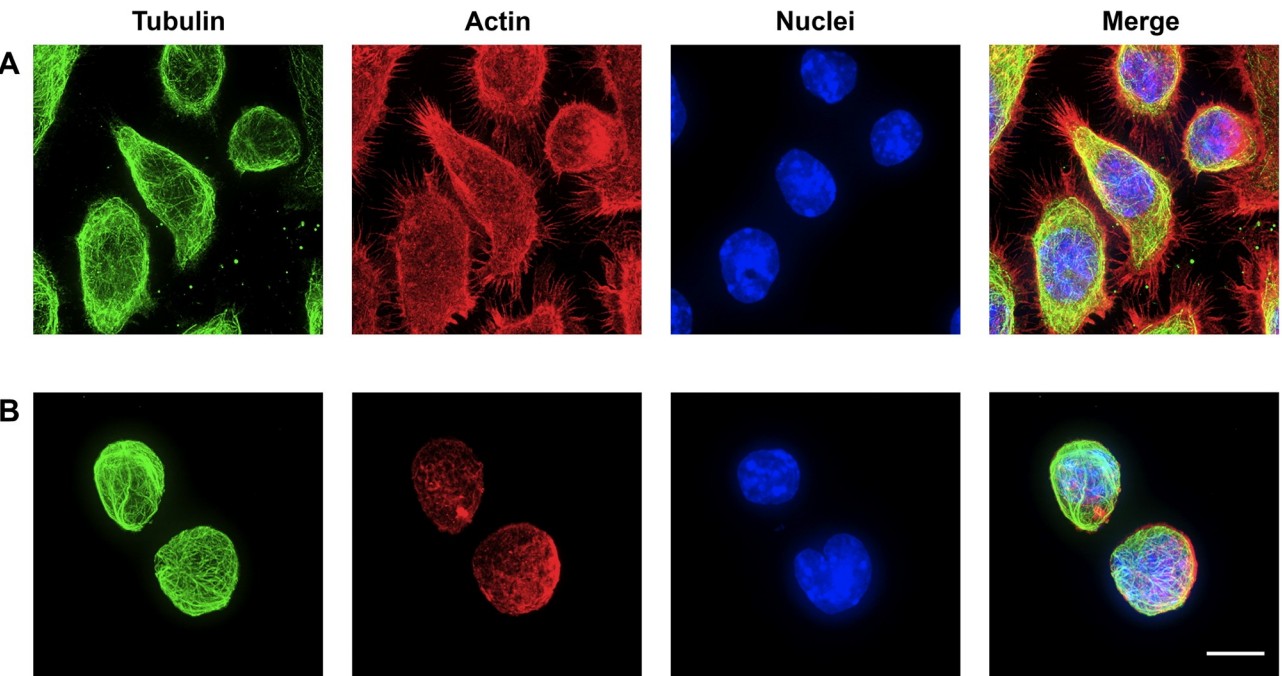

**Fig 2. Analysis of the effect of adhesion on the morphology of the actin and tubulin networks.** The actin and tubulin networks are visualized by confocal microscopy as described in the material and methods section, on adherent RAW264.7 cells (row A) and non-adherent RAW 264.7cells (row B). The contrast between the basket-like shape for the microtubule network in non-adherent cells and the more spider web-like shape in adherent cells is noteworthy, as well as the reorganization of the actin network. Scale bar: 20μm.

## Effects on the cytoskeleton

As expected, many proteins involved in the control of the cytoskeleton were highlighted in the proteomic screen, which was further illustrated by the "cell-cell adhesion" and "actin binding" clusters found in the pathway analysis. Such proteins for which a change in abundance was detected in at least one protein form included arp2 (P61161), arpc2 (Q9CVB6), capg (P24452), caza2 (P47754) cofilin 1 (P18760), dynamin 2 (P39054), GDIB (Q61598), gelsolin (P13020), GIPC1 (Q9Z0G0), lsp1 (P19973), memo1 (Q91VH6), gamma parvin (Q9ERD8), plastin-2 (Q61233) and beta tubulin itself (P99024).

To validate these findings further, we performed a microscopic analysis of the tubulin and actin cytoskeleton. As shown in Fig 2, the network of microtubules, which extends like a spider's web when the cells are adherent (Fig 2A), reorganizes into a basket net-like network that extends into the thin cytoplasmic space surrounding the nucleus. In this latter state, many microtubules appear to form bundles (Fig 2B).

The actin cytoskeleton of adherent cells shows rare/no stress fibers, but abundant filopodia and branched actin. The actin network is completely reorganized in non-adherent cells, with the disappearance of filopodia and the presence of a sub-cortical network.

## Phagocytosis and nitric oxide production

Phagocytosis, i.e. one of the important functions of macrophages, depends on the cytoskeleton, as shown by its well-known inhibition by cytochalasin, which blocks actin polymerisation [34]. In addition, GIPC1, which is one the proteins highlighted in the proteomic screen, is known to modulate phagocytosis [35]. Furthermore, the stiffness of the substrate to which the cells adhere has been described to modulate the differentiated functions of the macrophages

**Table 4. Assay of macrophages functions.**

| Condition | Adherent | Non-adherent |
|---|---|---|
| %phagocytic cells | 65.1±4.3 | 70.9±1.3 |
| Phagocytosis MFI * | 2536±203 | 2698±76 |
| LPS-induced NO production | 4.79±0.17 μM | 4.52±0.13 μM |
| LPS-induced IL-6 production | 8516±262 pg/ml | 8799±198 pg/ml |
| LPS-induced TNF production | 4311±461 pg/ml | 4241±143 pg/ml |

*: the mean fluorescence index (MFI), expressed in arbitrary fluorescence units, is an indicator of the number of fluorescent beads internalized by the cells, and thus of the capacity of phagocytic cells to internalize several beads.

[36]. Consequently, there could be a difference in these specialized functions between adherent and non-adherent cells. We thus measured the phagocytic capacity and the LPS-induced nitric oxide (NO), interleukin 6 (IL-6) and tumor necrosis factor alpha (TNF) production for cells in the adherent and non-adherent state. The results, displayed in Table 4, did not indicate significant differences in functionalities between adherent and non-adherent cells.

## Discussion

Adherence to a substrate is a requisite for most cells types, and cells are able to adopt various shapes to adapt to the adhesive characteristics of their substrate [37]. Nevertheless, there are a few situations, such as the metastatic process or blood cell diapedesis, where cells alternate between adherent and non-adherent behaviors. In order to test the changes that occur between these two states, a system is needed where cell physiology is not too altered between the adherent and non-adherent states, so as not to confuse changes in cell physiology with changes in adhesion status. Macrophage cell lines are among the models that allow such studies. Although they are known to respond to the characteristics of their substrate when they adhere [36, 38], their differentiated functions such as phagocytosis and inflammatory response to LPS are equivalent between an adherent state on plastic and a non-adherent state.

We thus used this model to study the changes linked to the adherence status, using a proteomic screen to get a wider appraisal of the phenomena at play. To this purpose, we chose to perform 2D gel-based proteomics, as this proteomic setup is able to detect changes in posttranslationally modified protein forms [39] and thus get closer to the cell physiology [40]. In the same trend, we chose to validate the proteomic results by functional assays rather than assays based on protein amounts [41]. Furthermore, in the case of 2D gel-based proteomics, a significant change in a modified form of a protein, which can be biologically significant because of the importance of post-translational modifications, can be revealed while the bulk of the protein remains constant, which would mask the change of interest in a protein amount-based validation.

The first (and expected) class of proteins that were found modulated corresponded to proteins playing a role in the cytoskeleton architecture. Several proteins implicated in the actin cytoskeleton regulation were modulated. Among them, several changes were detected at the post translational level as indicated by changes observed for acidic, modified forms of the proteins while the bulk of the proteins, as measured by the sum of the various protein spots for the same protein, did no change (e.g. arpc2 (Q9CVB6), capg (P24452), cofilin (P18760), GDIB (Q61598)). Furthermore, a change in the amount of gamma parvin (Q9ERD8) was detected. This protein belongs to the actinin superfamily [42] and plays a role in leukocyte adhesion [43]. Consistent with its role, gamma parvin was found in higher amounts in adherent cells.

Indirect regulators of the microtubule cytoskeleton were also found modulated, such as Memo-1 [44], pdcd 5 [45] or gpic, which is known to interact with integrins alpha-5 and 6 [46], integrins being known in turn to play a role in the architecture of the microtubules network [47]. Furthermore, a change in one beta tubulin spot was also detected. In view of what is known of the complexity of post-translational modifications for tubulin [48], and of the sequence similarities in the tubulin multigene family, a single spot assigned to one tubulin gene can be expected to be a mixture of several tubulin variants if not also products of different tubulin genes. It is therefore highly likely that the detected changes reflect a modification in the post-translational landscape of tubulin and not an overall change in the tubulin amount.

In line with all these changes, the architecture of the actin filaments and microtubules were found to be different between the adherent and non-adherent state, as expected. Regarding the actin network, the increase in the expression of gelsolin, a globular actin binding protein with severing activity, the actin-binding protein plastin-2, observed in cells in suspension (S1 Table), as well as the increase in the amount of S3-phosphorylated cofilin in the adherent cells, leading to a decrease of the actin severing activity in this condition, are most likely related to the rearrangement of the actin network that was observed.

Although gaining details in the mechanisms by which the cytoskeleton is modulated between the adherent and non-adherent states is of interest, an important added value of the proteomic screen lies in the less expectable pathways found modulated between the adherent and non-adherent states. A good example, still linked in some aspects to cell shape, is represented by the annexins. As in the case of cytoskeleton-associated proteins, the observed changes occurred in only one of the gel spots, generally an acidic, and thus modified variant. This may affect the fate or function of the protein, as described for annexin A1, where phosphorylation is associated with protein secretion [49], implicated itself in immune modulation [50]. Annexins are generally associated with cell membrane stability [51]. This can be on endosomal membranes, For instance annexin A1 regulates endosomal membranes stability [52], while annexin A4 is involved in plasma membrane curvature [53].

Another good example of rather unexpected results provided by the proteomic analysis is represented by the higher mitochondrial activity in the non-adherent state. This result is in sharp contrast with the loss of respiration observed upon cell detachment [54]. However, in this example, the respiration loss occurs in cells that are subject to massive death if left detached. Thus, this increase in mitochondrial activity may represent a survival mechanism to resist detachment. As metastatic cells are also an example of cells surviving to detachment, targeting the increase in mitochondrial activity may represent a putative therapeutic opportunity to fight metastatic cells. This change in the mitochondrial status is further supported by the increased abundance of the two mitochondrial antioxidant proteins prx3 and prx5 in the non-adherent state.

Beside mitochondrial activity, our proteomic and enzyme activity data pointed to changes in the central metabolism. Once again most of these changes occur on only a protein spot among several spots representing the same protein (e.g. pyruvate kinase, triose phosphate isomerase, enolase or phosphoglycerate mutase).

Changes in the central metabolism have also been observed in the epithelial-mesenchymal transition (EMT) [55]. In our case there was not a uniform trend for the enzyme activities changes. However, it may be worth noting that the two significant increases concern enzymes that catalyze the first step toward substrate utilization, namely hexokinase for glucose and acadl for lipids. Early enzymes in pathways are often controlling steps, as shown for hexokinase [56] and acadl [57], so that our findings may also correspond to a higher metabolic rate, as observed in EMT.

Finally, it is intriguing to find proteins associated with tumor progression in this proteomic screen, and with a mixed trend. Both proteins whose high level is associated with tumorigenesis and metastasis such as gipc1 [58, 59] or hyou1 [60] and proteins associated with tumor suppression such as vma5a [61] or pdcd5 [62] are found higher in abundance in non-adherent cells. Annexin A3, which has also been correlated with tumorigenesis [63], is found at a higher abundance in adherent cells. Although it must be kept in mind that the cells used in this study are cell lines, and thus cancer cells, they keep highly differentiated properties whether adherent or not. The mixed trend observed may represent a balance by which the cells keep their properties. In this respect, the situation investigated in the present study is radically different from EMT, where the cells change fate. Interestingly, when comparing the results obtained in the present study with those obtained by a proteomic investigation of EMT [55], the results were similar for the cytoskeleton associated proteins (e.g. arp2, arpc2, capg) and for other proteins such as vigilin or wdr1. However, the modulations of the mitochondrial proteins and of the tumor–associated proteins found in the present study were not found in the study on EMT, except for annexin A3 found higher in the epithelial state in [55] and higher in the adherent state in the present study. This argues in favor of a balance-keeping mechanism at play in the present case and not in EMT.

## Supporting information

**S1 Fig. Raw 2D gel images, adherent cells.**
(JPG)

**S2 Fig. Raw 2D gel images, non-adherent cells.**
(JPG)

**S3 Fig. Complementary annotated 2D gel images.**
(JPG)

**S1 Table. Raw results from the quantitative analysis of the 2D gels.**
(PDF)

**S2 Table. Proteins showing a significant change in abundance between adherent and non-adherent cells.**
(PDF)

**S3 Table. Modulated pathways highlighted by the DAVID annotation tool.**
(PDF)

**S4 Table. Semi-quantitative peptide analysis by spectral counting in the mono-phosphorylated form of cofilin.**
(PDF)

**S1 File. Cover supplementary material.**
(PDF)

## Acknowledgments

On a scientific level, the authors would like to thank the curators of the Swissprot database for the quality of their functional annotations, which made the exploitation of the proteomic data much more fruitful and straightforward.

## Author Contributions

**Conceptualization:** Laurence Lafanechère, Thierry Rabilloud.

**Formal analysis:** Sacnite Ramirez Rios, Anaelle Torres, Hélène Diemer, Sarah Cianférani, Laurence Lafanechère, Thierry Rabilloud.

**Funding acquisition:** Sarah Cianférani, Thierry Rabilloud.

**Investigation:** Sacnite Ramirez Rios, Anaelle Torres, Hélène Diemer, Véronique Collin-Faure, Thierry Rabilloud.

**Methodology:** Sacnite Ramirez Rios, Anaelle Torres, Hélène Diemer, Véronique Collin-Faure.

**Project administration:** Laurence Lafanechère.

**Supervision:** Laurence Lafanechère, Thierry Rabilloud.

**Visualization:** Anaelle Torres, Hélène Diemer.

**Writing – original draft:** Laurence Lafanechère, Thierry Rabilloud.

**Writing – review & editing:** Sacnite Ramirez Rios, Anaelle Torres, Hélène Diemer, Véronique Collin-Faure, Sarah Cianférani, Laurence Lafanechère, Thierry Rabilloud.

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
