## [Decision Letter · Decision Letter 0]

1 Mar 2021

PONE-D-20-35588

A proteomic-informed view of the changes induced by loss of cellular adherence: the example of mouse macrophages

PLOS ONE

Dear Dr. Rabilloud,

Thank you for submitting your manuscript to PLOS ONE. As you can see in viewing the reviewers comments, there is a wide spread of reviewer concerns, ranging from technical and informational to fundamental (in this case questioning the rationale and core experimental design of the cell adherence experiment).  In these instances I like to provide the authors with at least one round for careful consideration of all the critics and reply accordingly.  Therefore, we invite you to submit a revised version of the manuscript that addresses the points raised during the review process.

We look forward to receiving your revised manuscript.

Kind regards,

Jon M. Jacobs, Ph.D.

Academic Editor

PLOS ONE

Journal Requirements:

'This work was funded by the CNRS and the University Grenoble-Alpes. This work used the platforms of the French Proteomic Infrastructure (ProFI) project (grant ANR-10-INBS-08-03).

'SC-grant ANR-10-INBS-08-03, from Agence Nationale pour la Recherche, France.

The funder had no role in study design, data collection and analysis, decision to publish, or preparation of the manuscript'

Reviewers' comments:

Reviewer's Responses to Questions

**Comments to the Author**

1. Is the manuscript technically sound, and do the data support the conclusions?

Reviewer #1: No

Reviewer #2: Partly

Reviewer #3: Yes

2. Has the statistical analysis been performed appropriately and rigorously? 

Reviewer #1: No

Reviewer #2: Yes

Reviewer #3: Yes

3. Have the authors made all data underlying the findings in their manuscript fully available?

Reviewer #1: Yes

Reviewer #2: Yes

Reviewer #3: Yes

4. Is the manuscript presented in an intelligible fashion and written in standard English?

Reviewer #1: No

Reviewer #2: Yes

Reviewer #3: Yes

5. Review Comments to the Author

Reviewer #1: The study by Ramirez-Rios et al., addresses the proteomic changes observed in a mouse monocytic cell lines, RAW264.7 upon loss of adherence. This cell line grows in a mixture of adherent and non-adherent cells, thus this effort represents an attempt at understanding the molecular differences that mediate this distinct property. The authors use bi-dimensional electrophoresis to identify differential spots between lysates of adherent vs. non-adherent cells, which then they identify by mass spectrometry. Cluster analysis reveals differences mainly at cytoskeletal and adhesive proteins. Other interesting clusters emerge, e.g. mitochondrial proteins and metabolic enzymes. Next, they evaluate the enzymatic activity of some metabolic enzymes which they have identify as differentially expressed in adherent vs. non-adherent cells. Most differences, although significant, are modest. Likewise, they observe differences in mitochondrial proteins, but these do not translate into a significant difference in glucose consumption, although the mitochondrial transmembrane potential seems elevated in the non-adherent condition. Finally, they observe changes in cytoskeletal proteins, which are reflected in their intracellular organization in adherent vs. non-adherent cells. However, at a functional level, there is no fundamental difference between adherent and non-adherent cells in terms of phagocytosis and LPS-induced responses, which is a somewhat flat coda for the experimental part.

This study has a major issue, which is the feeling that the initial hypothesis is somewhat weak. Some of the results are striking. Adherent vs. non-adherent cells are morphologically different, yet they both perform their phagocytic function well and respond to stimuli in comparable fashion. I understand that cells in culture are somewhat heterogeneous and a population at a given time is made of cells in different phases of the cell cycle, are of different generation, etc. In essence, are the authors sure that the whole story is not a culture artifact. Additional proof is needed at the gate, including convincing answers to the following questions: Are non-adherent cells and adherent cells fundamentally different populations? Or just cells on a different part of their cycle? If the authors take only the adherent cells and re-seed them, do they obtain non-adherent cells? Or all the cells remain adherent? Same with non-adherent cells, if they separate these and culture them separate from the adherent population, do some of them become adherent? The difference in the proteomics data, although promising of intrinsic heterogeneity, fails to reveal whether the adherent vs. non-adherent cells are indeed different populations, or different phases of the same population. In this regard, if quantifications hold, the proteomics differences appear promising, but the rest of the study fails to identify a solid premise to explain the differences in adherence. Furthermore, the (relatively) negative results in terms of mitochondrial function and very modest metabolic differences actually cast doubt on the significance of the proteomics data.

Other specific points stand out:

1. Table 1 does not provide an idea of the relative abundance of a given protein in one condition and the other. I assume that relative abundance is determined based on peptide counts, but this is not clear. Thus, the differences can be misleading in the absence of a score. Hence, data in Table 1 needs a sort of quantitative index to determine the degree of difference. A volcano plot would probably be more informative.

2. The modest differences in metabolic enzymes in Table 2 are surprising, because one would assume that major morphological differences cause a generally different metabolic rate. At any case, it is surprising that the authors have not chased this more vigorously.

3. Cytoskeletal organization differences are intrinsic to the fact that cells are adherent or non-adherent, so Figure 2 does not make a whole lot of sense as is.

4. It is surprising that the authors have not measured the major parameter that defines metabolic usage, which is cell division. Do adherent and non-adherent cells divide differently? For the answer to this question to be significant, they need to address the questions in the main body of text.

Reviewer #2: Comments to the Author

In this manuscript, Sacnicte et.al. have make a comparative analysis of the transition between adherent and non-adherent states of RAW264.7 cell line through a proteomic approach. They showed that the metabolic enzymes and mitochondrial activity was increased in non-adherent cells compared with adherent cells, without differences in glucose consumption and functions of macrophages. I would have a couple of suggestions:

MAJOR

1. Protein levels of highlighted changed metabolic enzymes described in the proteomic data, as ACADL, HK3, PGAM and KPYM should be demonstrated, e.g. by WB.

2. Protein levels of several mitochondrial proteins described in the proteomic data, should be demonstrated as well.

WHOLE MANUSCRIPT

1. Line 33 and elsewhere: Change esperiments to experiments.

Reviewer #3: This paper describes a 2D electrophoresis study that found differentially expression protein bands in the gels and identified those bands with mass spectrometry. The system studies was a cell culture system of adherent and non-adherent cells. The change in phenotype between these two states was also explore with a series of biochemical assays and imaging studies.

The experiments described in the paper are all technically sound. The results do provide new information about altered biochemical processes between the two states.

The primary limitations of the paper are, 1) the use of a single adherence model in mouse, and 2) the use of 2D electrophoresis to determine the changes in protein expression.

I would have liked to see at least some basic details about the mass spectrometry experiments, particularly the instrument used. I think it would also be appropriate to be clear about how many bands in the gels had statistically significant changes in expression and how many of those proteins were identified.

6. PLOS authors have the option to publish the peer review history of their article (what does this mean?). If published, this will include your full peer review and any attached files.

Reviewer #1: No

Reviewer #2: No

Reviewer #3: No

---

## [Author Response · Author response to Decision Letter 0]

26 Apr 2021

First and foremost, we would like to thank warmly the editor and the reviewers for the time and effort that they have devoted to our manuscript. Their input has been very precious to us, and we hope that this revised version will meet their expectations.

Please note that for helping the reviewers to easily find the major changes in the manuscript, the added text should appear in blue in the revised version. We have also added page and line numbering.

Reviewer #1

This study has a major issue, which is the feeling that the initial hypothesis is somewhat weak. Some of the results are striking. Adherent vs. non-adherent cells are morphologically different, yet they both perform their phagocytic function well and respond to stimuli in comparable fashion. I understand that cells in culture are somewhat heterogeneous and a population at a given time is made of cells in different phases of the cell cycle, are of different generation, etc. In essence, are the authors sure that the whole story is not a culture artifact. Additional proof is needed at the gate, including convincing answers to the following questions: Are non-adherent cells and adherent cells fundamentally different populations? Or just cells on a different part of their cycle? If the authors take only the adherent cells and re-seed them, do they obtain non-adherent cells? Or all the cells remain adherent? Same with non-adherent cells, if they separate these and culture them separate from the adherent population, do some of them become adherent?...and

4. It is surprising that the authors have not measured the major parameter that defines metabolic usage, which is cell division. Do adherent and non-adherent cells divide differently? For the answer to this question to be significant, they need to address the questions in the main body of text.

Reply

It is completely true that what was obvious for us who grow these cells routinely was far from obvious in the text. To address the reviewer’s concerns, we have performed two sets of experiments. 

In the first set of experiments, we have taken cells that are grown under our routine conditions (i.e. on non-adherent plastics), seeded them on non-adherent or adherent plastics and let them grow for 24 hours. We then just enumerated the number of non-adherent cells in both conditions. While nearly all cells remained non-adherent on non-adherent plastics, close to 95% of the seeded cells were no longer in suspension when seeded on adherent plastics. In our opinion, this rules out any population bias. The reverse experiment was not performed, because cells cultured on adherent plastics are difficult to recover and thus show variable mortality from one experiment to another. This would lead to artefactual (dead or poorly viable) cells remaining in the supernatant and biasing the cell enumeration.

In the second set of experiments, we analyzed the cell cycle under both conditions (adherent vs non-adherent). No significant differences could be found. These experiments are described in lines 81-85 and 109-140 for the methods, and 253-267 for the results. 

In our opinion, this shows that when cells adhere, they alter their proteome, leading to the changes that we observed. 

Reviewer #1

1. Table 1 does not provide an idea of the relative abundance of a given protein in one condition and the other. I assume that relative abundance is determined based on peptide counts, but this is not clear. Thus, the differences can be misleading in the absence of a score. Hence, data in Table 1 needs a sort of quantitative index to determine the degree of difference. A volcano plot would probably be more informative.

Reply :

In 2D gel-based proteomics, the quantitative analysis is NOT carried out in the mass spectrometer, but on the 2D gel images at the protein staining level, and not at the peptide level. Thus, when the gels are analyzed, not all the spots are statistically different (of course) and opposite to what happens in shotgun proteomics, not all spots are analyzed by mass spectrometry. Mass spectrometry analysis is restricted only to spots that do show a significant change between conditions of interest. For the sake of completeness of data, we have now provided the complete gel analysis table (S1 Table). This will allow the interested readers to perform statistical analyses on the raw data.

Reviewer #1

2. The modest differences in metabolic enzymes in Table 2 are surprising, because one would assume that major morphological differences cause a generally different metabolic rate. At any case, it is surprising that the authors have not chased this more vigorously.

Reply

This is exactly why we cross-checked the proteomic results by enzyme activities. The outcome is that the differences are subtle. We agree that this may seem surprising, but these are the real data. They are also consistent with the weak differences observed in the cell cycle experiments

Reviewer #1

3. Cytoskeletal organization differences are intrinsic to the fact that cells are adherent or non-adherent, so Figure 2 does not make a whole lot of sense as is.

Reply

It is completely true that cytoskeletal differences are intrinsic to the fact that cells are adherent or non-adherent. However the interest of figure 2 is to document what the actual differences are, and this is why we are inclined to keep it. 

Reviewer #2: 

MAJOR 

1. Protein levels of highlighted changed metabolic enzymes described in the proteomic data, as ACADL, HK3, PGAM and KPYM should be demonstrated, e.g. by WB. 

2. Protein levels of several mitochondrial proteins described in the proteomic data, should be demonstrated as well.

Reply

We chose not to perform such validation by WB, because the real strength of 2D gel-based proteomics is to highlight changes in modified protein forms that can be functionally important but not detectable at the bulk protein level, which is what WB probes. These reasons are now detailed in lines 442-451 and 505-507

Reviewer #3: 

The primary limitations of the paper are, 1) the use of a single adherence model in mouse, and 2) the use of 2D electrophoresis to determine the changes in protein expression. I would have liked to see at least some basic details about the mass spectrometry experiments, particularly the instrument used. I think it would also be appropriate to be clear about how many bands in the gels had statistically significant changes in expression and how many of those proteins were identified.

Reply to 1)

There are not that many models that allow to probe adherence-linked changes without changes in viability, proliferation or differentiation status. So we are happy to report data on one of these rare models

Reply to 2)

as stated above in the reply to Reviewer 2, the real strength of 2D gel-based proteomics is to highlight changes in modified protein forms that can be functionally important but not detectable at the bulk protein level, as detected for example by shotgun proteomics. This is now discussed in lines EEEE.

Regarding the details of the mass spectrometry analysis, they are now added in lines 146-163

Regarding the numbers of spots showing statistically significant changes vs. number of them identified, this information now appears in lines 279-282

---

## [Editor Report · Decision Letter 1]

17 May 2021

A proteomic-informed view of the changes induced by loss of cellular adherence: the example of mouse macrophages

PONE-D-20-35588R1

Dear Dr. Rabilloud,

We’re pleased to inform you that your manuscript has been judged scientifically suitable for publication and will be formally accepted for publication once it meets all outstanding technical requirements.

Kind regards,

Jon M. Jacobs, Ph.D.

Academic Editor

PLOS ONE
---

## [Editor Report · Acceptance letter]

20 May 2021

PONE-D-20-35588R1 

A proteomic-informed view of the changes induced by loss of cellular adherence: the example of mouse macrophages 

Dear Dr. Rabilloud:

I'm pleased to inform you that your manuscript has been deemed suitable for publication in PLOS ONE. Congratulations! Your manuscript is now with our production department. 

Kind regards, 

on behalf of

Dr Jon M. Jacobs 

Academic Editor

PLOS ONE